# The evolution of the role of nursing in primary health care using Bourdieu's concept of habitus. A grounded theory study

Cristina Blanco-Fraile[1], María Madrazo-Pérez[1], Victor Fradejas-Sastre [1,2] *, Esperanza Rayón-Valpuesta[3]

**1** Faculty of Nursing, University of Cantabria, Santander, Cantabria, Spain, **2** Hospital Universitario Marqués de Valdecilla, Santander, Cantabria, Spain, **3** Faculty of Nursing, University of Complutense of Madrid, Madrid, Spain

* victor.fradejas@unican.es

**Data Availability Statement:** All relevant data are within the paper. Supporting Information Files are available on request from the University of

## Abstract

### Aims

To analyse the global process by which Spanish nurses have acquired a differentiated role in primary health care and to develop a theory that explains the evolution of this role.

### Design

Grounded Theory was selected, as proposed by Glaser and Strauss, following the theoretical framework of Bourdieu's *habitus*.

### Methods

Thirteen in-depth interviews were conducted between 2012 and 2015, using theoretical sampling and seeking maximum variability. The analysis of the data included progressive coding and categorization, constant comparative analysis and memo writing.

### Results

A core category emerged, "Autonomy", composed of three categories: "Between illusion and ignorance. Genesis of a *habitus*", "The recognisable and recognised *habitus*" and "*Habitus* called into question", showing the genesis of the nursing role in primary health care and the elements that influence the autonomy of the role: the ability to decide their training, assume their own leadership, configure teams and acquire independent skills. "Seeking autonomy" was the substantive theory that emerged from the data.

### Conclusion

The results reveal the elements that strengthen the autonomous professional role and that this role is legitimated when two elements are identified: the acquisition of a *habitus*, based on practices carried out regularly and the recognition of this *habitus* by the population and others professionals.

Cantabria Archive (http://repositorio.unican.es/) for researchers who meet the criteria for access to confidential data. This restriction is due to ethical compliance in order to not compromise study participants' privacy, since the dataset derives from clinical studies involving human participants. Relevant data Requests may be sent to Carmen Chasco Vila, Responsable de División de Medicina y Enfermería. Biblioteca Universitaria. Facultad de Medicina. (Telephone number: + 34 942 201 217; E-mail address: carmen.chasco@unican.es).

**Funding:** The authors received no specific funding for this work.

**Competing interests:** The authors have declared that no competing interests exist.

## Impact

The results of this study identify the elements that guide and strengthen the professional role and redefine the concept of autonomy. These are operational findings and could potentially be used to define new strategies for advancing the role of nursing in primary health care.

## Introduction

At the Alma Ata Conference in 1978, the World Health Organisation [1,2] pointed out the importance of primary health care (PHC) as the first level of health service delivery. In general terms, PHC was defined as a new model of health care worldwide [2,3] with great achievements in health care, in which nursing plays a relevant role in the achievement of its objectives [4] and has even been identified as the centre of PHC [5].

This change of model has forced a new definition of professional roles [6–8]. Both nurses and doctors have had to develop new skills and competencies, thus configuring a complex professional context in which forces, conflicts and monopolies have been generated [3,9,10].

The implementation of the PHC model in Spain in the 1980s, has enabled the nursing professionals, all of whom are generalist nurses (RN), to participate with an equal level of responsibility compared to other professionals [11]. Although Spanish nurses working in PHC have achieved many accomplishments, to the point of being proposed as a model [12], there is still a certain lack of definition concerning the role and new challenges to be addressed [13].

The arrival of the new professionals, both RNs and doctors, was conditioned by the dominant models taken from hospital care, with a focus on pathology and with a tendency towards curative care, in accordance with their training oriented to addressing the immediate patient needs [14]. This has meant that throughout these almost 40 years, the RN role in PHC has fundamentally been built based on practice, i.e., learning by doing. The new role of nursing in PHC was proposed under broad, unspecific lines, however, the population's need for professional care was growing exponentially and was not only based on the immediacy of care. In the absence of a model or reference guide, RNs have been constructing the new role as they work, based on trial and error. To be able to analyse this role and overcome these new challenges, the present study aims to determine the process, from a holistic viewpoint, through which Spanish nurses have developed a differentiated role in PHC.

### Background

There is extensive research on nursing roles in PHC, on both RNs and advanced practice registered nurses (APRNs). Several authors point out the lack of knowledge, inadequate assessment or lack of understanding of the nursing role in PHC [15,16] or the ambiguity and fragility of the role [17].

In Spain, before the 1980s, there were two types of care provided outside the hospital setting: in outpatient clinics and in rural areas [18]. In outpatient clinics, consultations were largely devoted to administrative procedures, such as obtaining sick leave, tasks that were often delegated to the nurse who accompanied the doctor in the consultation. Sick people were only attended to when they came to the doctor. In rural areas, care was provided at home. In both cases there was a big difference with the hospital environment, which was more financially endowed and more prestigious. Moreover, the nurses were all RNs, as there was no APRN and the specialty of Community Nursing was approved in the year 2012.

Specifically, the role of Spanish nurses in PHC has been studied with the focus on the analysis and identification of the interventions that PHC nurses carry out, highlighting the follow-up of patients with chronic disease, home care and promotion and prevention tasks [19,20].

Most researchers suggest that the challenges of PHC in Spain are focused on the assumption of new skills, according to the patient presentations [19,21]. Other authors describe the difficulties involved in meeting this challenge, [20,22].

However, in the study of the professional role, it is fundamental to know where one starts from and what path one has followed. On an international level, although the origin of PHC is known [3], there is no evidence of work that has analysed the process by which a new role has been configured, which is essential to design future strategies, given that there is still, a need to clarify this role [23,24].

To analyse this process, our study adopted Bourdieu's sociology as a theoretical framework, which has much to offer as a reference for nursing research and allows us to understand central aspects of the professional role [25], especially through the concept of *habitus* [26].

"The *habitus* is this kind of practical sense for what is to be done in a given situation" [27]. It is understood as a system of social competencies that indicate the "ability to" do something, together with the "social recognition" to exercise those competencies regularly. The *habitus*, therefore, has special relevance in practice, and therefore, in the development of the professional role, because as a system of dispositions to a certain practice, it generates regular behaviours [28].

The *habitus*, in addition, presents a useful potential for predicting behaviour by possessing a certain prognostic capacity. This means that by knowing the *habitus*, it is possible to somewhat predict future practices, since the *habitus* makes social agents that are subjected to the same circumstances tend towards the same response [29].

## The study

### Aims

To analyse the global process by which Spanish nurses have acquired a differentiated role in PHC and to develop a theory that explains the evolution of this role, from its beginnings.

In this study, the following research question was addressed: what process has taken place in the development of the professional role of Spanish PHC nurses, since the inception of the model in 1985?

### Design

The choice of design should make it possible to discover, through one' s own experiences, what duties and responsibilities nurses have fulfilled over the years, and why they have done so, until their professional role has been established.

For this purpose, Grounded Theory was selected [30,31] including progressive coding and categorization, constant comparative analysis, theoretical sampling, and memos.

### Sample/Participants

The sample was conformed of nursing professionals (both staff nurses and nurse managers) who participated in the development of the new PHC model in the region of Cantabria (Spain), some of them from its inception.

The selection of participants was first made by intentional sampling and then continued with theoretical sampling, with the purpose of finding the maximum variability of the sample [31] until saturation was reached. The first nurse manager and two other later directors from

different periods all of whom were interviewed to collect testimonies from managers who had joined the model at different points in its evolution since 1986. Most of the participants had over 10 years of experience in PHC.

The inclusion criteria were: voluntary participation in the study, being active, i.e. still working in PHC at the time of the interview; some for a long period of time in the different stages of PHC since its origin, and who would perform both staff nurse and nurse manager roles.

The first informants were accessed through the current Director of Nursing in PHC, through a letter of invitation, followed by a phone call.

## Data collection

In-depth interviews were the technique of choice for data collection. The mean duration was 90 minutes. A total of 13 interviews were conducted with 12 informants. They were recorded after obtaining informed consent and later, transcribed verbatim. Open-ended questions were asked, following a general thematic outline presented in Table 1.

The interviews were conducted by the principal investigator, who at the time was following a doctoral program and who had experience as a PHC nurse. She did not previously know the participants. After each interview, simultaneously to the transcription, the suggested ideas were written, and analytical memos were elaborated.

Data collection began with a first interview with an informant who participated in the first primary care team as a nurse and who held management responsibilities. This interview enabled a first approach to the research topic through the open question "What went on here?" She was considered a key informant because of her privileged position at the origin of the process, so a second interview was held to clarify possible doubts, confirm the data, and refine the interpretation. Data collection and analysis was conducted from July 2012 to June 2015.

## Ethical considerations

This research was conducted according to the Declaration of Helsinki [32]. The informed consent of all the participants included in the study was requested and provided in writing, informing participants of the voluntary nature of their participation and the possibility of withdrawing from the study at any time. The data obtained were treated to guarantee confidentiality and anonymity, according to the Law on Protection of Personal Data in force in Spain [33].

**Table 1. Interview script (English).**

| |
|---|
| • Arrival: How did you arrive at PHC? |
| • Training: upon arrival and over time |
| • Competencies: What work did you do? |
| • Differences and similarities with previous work experiences |
| • Relationships with other nurses |
| • Relations with directors |
| • Interprofessional relations: What was the working environment like? |
| • Services offered to the population: What were the services offered, what kind of activities were carried out? |
| • Important / critical moments |
| • Characteristics of Nursing Work |
| • Nursing contributions: what do you think they are? |
| • Future perspectives |

## Data analysis

Following the recommendations of Morse [34], a detailed description of the complex process of analysis and coding carried out is presented in Table 2.

After analysing the first interview with the key informant, a series of categories were extracted that served to elaborate the script for the following interviews, thereafter the interviews and the corresponding analyses were alternated; even some properties and relationships

**Table 2. Process of analysis, codification and categorisation.** Central category.

| PHASES | PROCESS | RESULTS OF THE PROCESS IN EACH PHASE / EXAMPLES | RESEARCHERS |
|---|---|---|---|
| PHASE 1. OPEN CODING | • Verbatim transcription of the recorded interviews<br>• Line-by-line reading<br>• Identification of words or expressions rich in meaning (through sensitivity)<br>• Code assignment<br>• Classification of codes according to level of abstraction, from simpler to more complex by constant comparison.<br>• Grouping of codes by similarity of meaning and construction of broader categories or subcategories<br>• Elaboration of analytical memos | • A high number of codes were obtained.<br>• Identification of new concepts to be explored in subsequent interviews (rewriting of the interview script)<br>• Search for new informants (theoretical sampling)<br>• Example 1: "that patients believed in nurses": SEARCH FOR CREDIBILITY<br>• Example 2: "we were naked but with illusion"; "I started from scratch": NEED FOR TRAINING / ILLUSION, EXPECTATIONS<br>• Example 3: "the doctors didn't want their nurse to do. . ."; "we suffered a lot": FIGHTING / SIGNS OF CONFLICT<br>Example 4: "give in to grow"; "find a space": IT IS NECESSARY TO ESTABLISH NEGOTIATION | • Principal investigator<br>• Discussed with the other researchers on the team to confirm coding and categorization, especially for complex or questionable concepts |
| PHASE 2. AXIAL CODING | • Comparison between subcategories<br>• Relationship and axial comparison (crossover) between subcategories and construction of new superior subcategories<br>• Drawing up analytical memos | Construction of new higher subcategories (higher level of abstraction)<br>Identification of new concepts to be explored in new interviews<br>Search for new informants (theoretical sampling)<br>Example:<br>*credibility + training: LEGITIMATION<br>*illusion + expectations + struggle: INITIATIVE<br>* conflict + struggle + initiative: CONQUEST<br>* initiative + legitimation + conquest: RECOGNITION / INDEPENDENCE | • Principal investigator<br>• Discussed and advised with other researchers of the team, proposing different alternatives of categorization and axial relations |
| PHASE 3: SELECTIVE CODING | • Constant comparison between increasingly broad and abstract subcategories to reach the 3 main categories.<br>• Emergence of the central category AUTONOMY<br>• Analysis and discovery of autonomy, implicitly or explicitly throughout the process. Confirmation. Definition of the theory: "Searching for autonomy" | • Main categories emerge<br>• Example: **legitimisation + conquest + recognition + independence**: serves to build, along with other forms of categorization, one of the main categories: THE RECOGNISABLE AND RECOGNISED HABITUS | • Principal investigator<br>• Discussed and advised with the entire research team seeking limitations, and differences between the research group reaching a final consensus. |
| PHASE 4: CENTRAL CATEGORY | • Identification of the transversal concept present in the main categories<br>• Discussion and verification of the proposal (confirmability)<br>• Discussion of the term | AUTONOMY | • Principal investigator<br>• Discussion with key informant<br>• Discussed and analysed with the entire research team until the final result is reached. |
| PHASE 5: THEORY FORMULATION | • Development of possible statements and proposals that respond to the reality of the phenomenon (confirmability) | "SEEKING PROFESSIONAL AUTONOMY" | • Research team |

began to be discerned that were later confirmed as "provisional hypotheses" [31] supported by analytical memos.

The coding process was divided into three phases [31]: open, axial and selective coding.

The open coding was performed line by line, attributing a code or meaning to the informant's words [35] with the support of the Nvivo v.10 program (QSR International Pty Ltd, 2012), and by writing analytical memos.

The experience of the principal Investigator in PHC, was greatly useful to recognize patterns, similarities and differences [34], making the implicit explicit by applying theoretical sensitivity [30], which among other aspects, facilitated the decision to opt for Grounded Theory.

From the initial stages of this study the emerging concepts were ordered according to the timeline described by the informants, which made it possible to identify different periods of time, differentiated from each other according to the content of their narratives, and which were marking a change of era A theoretical framework was used in the form of a diagram, inspired by the conditional/consequential matrix [31] to explain the relationships of the concepts among each other for each time period.

## Validity and reliability/rigour

The proposals by Strauss and Corbin [31] and Calderón [36] have been considered following an inductive, orderly, and systematic process, from data to theory.

Credibility, understood as concordance with the phenomenon, was achieved by seeking maximum variability in the sample, conducting a second interview in the case of the key informant and then discussing the findings, thus facilitating interpretation. For this purpose, the results were also returned to the informants, that is, the results of the work were communicated to the nurse who conducted the pilot interview, as well as to the key informant, both of whom identified and confirmed the coherence (credibility) with their own reality. Also, through discussion within the research team, thereby facilitating researcher triangulation, which was applied during the coding and categorization process, and also in the interpretation of the results, through a second review by another researcher of the data proposed by the main researcher. Any aspects that were doubtful or divergent were further discussed within the team. The memos supported the continuous reflective analysis, both for the progressive selection of the sample and during the analysis process.

The usefulness and impact of the results for practice and for future research is developed below as recommendations for practice. Originality can be valued especially through the diachronic perspective in the analysis of the nursing role and, in addition, through two fundamental aspects: first, the application of Bourdieu's theoretical framework to the practice of health care, and second, the results obtained, which illustrate in an operative manner the elements that favor and those that hinder the development of the professional role.

Likewise, attempts were made to control possible biases through a continuous exercise of reflexivity, being aware that the position of the principal investigator could influence the results because of her previous experience in PHC.

## Results

Of the 12 people interviewed, nine were women and three were men. Of these, six had held the position of head nurse in their team at some point, three had worked in PHC since its inception in 1985, and two had held a position on a temporary basis. One of the informants had joined PHC after extensive experience in a hospital and another had previously held a position as a rural nurse. Table 3 illustrates complementary narratives by the study participants.

**Table 3. Additional participant narratives.**

| MAIN CATEGORY | CONCEPT | NARRATIVE |
|---|---|---|
| 1.- BETWEEN ILLUSION AND IGNORANCE. GENESIS OF A HABITUS. | Entering PHC | *Well, I experienced it as a new world because the little experience I had was performing replacements and then I knew how the old model worked, the old practices. . .*<br>Question: What led you to work in Primary Health Care?<br>*For me there was no particular vocation, the need to find a job.* (I4) |
| | Searching for teamwork | *We were all completely naked, but eager. . . So, of course, we were very much looking for teamwork.*(I1) |
| | Selecting a leader, beginning the "struggle", the illusion is born | *I was involved up to my neck. Exciting, difficult, many times in a struggle that I sometimes did not have arguments to defend, because I did not control the subject perfectly either. I mean, I was learning like everyone else, I didn't know any more than anyone else.*(I3) |
| | Horizontal and participatory leadership | *"Hey, and how do we do this?" maybe because we were all on the same level, we were more cohesive than they were* [the doctors].(I1) |
| | Nursing "consultation" begins | *[The consultation] meant that I had to "refresh" many things and it was an effort, but at the same time it was rewarding to have my own office and the fact that my work was not merely assistive.*(I4) |
| | Confrontations with doctors for wearing white lab coats | *What happens is that there were confrontations with doctors who did not want their nurse to wear a white lab coat like himself.*(I2) |
| | Patients were unaware of the new role, identifying the nurses based on their techniques | *And besides, the technique was very much identified by the patients, it was what the nurse did, because the patient didn't expect anything from the nurse herself: just that she would jab him with a needle. So, what did the patient expect from the nurse? In the beginning no one really knew.* (I2) |
| | Conquering the legitimacy of the new role | *So you had to win them over little by little, first the doctor and then the patient, because the patient didn't assume that you were going to take care of what was happening to him either.* (I8) |
| 2.- THE RECOGNISABLE AND RECOGNISED HABITUS | New competencies in PHC | *I have been discovering the work with the community, the work in community care, and it seemed to me that by collaborating, I was opening a niche and making the role of the nurse become an important element.*(I2) |
| | Expanding knowledge | *I signed up for all the courses there were in gynaecology, digestion, team building. . .*(I12) |
| | Reference for the population. Holistic vision | *I think we are a trusted reference for health problems of all kinds, not only for each patient but also for those around them.* (I6) |
| | Own role | *I think that the population does demand the work of nurses, it knows what we are for, now it knows that it wants to change doctors but it does not want to change nurses. . .*(I3) |
| | Strategies for minimizing conflict with physicians: dialogue | *It's just that if you take away our consultations, then there can't be any nursing work. Then they also reconsider.*(I1) |
| | Constitution of medical-nursing teams | *Then, for example, at the level of medical-nurse team and agreements, each team is a world. Each team works in a particular way.*(I5) |
| | Consolidation of the nursing team and leadership | *With nursing it's the same: you need a group, you need a team that functions as such, that is consolidated and for there to be leadership; and a group that works with that leader.*(I9) |
| | Identification of the autonomous role | *Because you are an autonomous professional, you do not just do what you are told, but you know how to do things autonomously. Recognition. . .I mean, before they sort of had the image of the nurse as the doctor's assistant and I think that's not the case now.*(I1) |
| | Official undefinition of the role. | *Well, we have never really had clear and defined functions. So, we have been taking up some space. We've been taking up space: "I'm taking this area of work, this is for me, I'm going to follow it.* (I4) |

*(Continued)*

**Table 3.** (Continued)

| MAIN CATEGORY | CONCEPT | NARRATIVE |
|---|---|---|
| 3.- HABITUS CALLED INTO QUESTION | Influence of political management on the role | *I have the feeling that we went from having a certain autonomy in Primary Health Care to having everything established from the management.*(I11) |
| | Instrumentalization of the service portfolio | *The portfolio of services has been created "just for the sake of appearances". I do not believe that there has been a serious and meditated programming.* (I4) |
| | Weakening of management and leadership | *I see the nurse's work as emerging, holding up a little bit, and starting to decline. This has to do with some managers who have never moved a finger to promote the work of nursing, and it has been quite a few years. . . .. What managers usually do is wipe the slate clean, and that can't be done.* (I3) |
| | Demographic factors affecting the role | *Lately you can see that people are getting older and I am quite in demand now. Well, look, say 40% of the consultation has been in demand, you can tell that people are getting older.* (I10) |
| | Effects of computerization | *We should be encouraging people to make care plans. . . but the [computer] tool must be easier, simpler. . . Not just fill in the blanks, but all the assessment that you already have integrated here, because you know the patient. . . The problem is that this program does not allow me to do that. My care as a nurse goes there in free text with minimal interventions.*(I2) |
| | Inequality in the performance of the role | *Now you continue working the same as you used to, but as a more independent person: you do it if you want to, the other person doesn't, so and so whatever. . .*(I8) |
| | The importance of evaluating the work of nurses in order to achieve results and visibility | *An obsession of mine is to evaluate health outcomes for everyone. Yes, you can. What you have to know is if you want to do it, but you can. Today, for example in PHC we have millions of data that can be transformed into information that can be used to make decisions based on the qualitative response you expect.*(I11) |

PHC: Primary Health Care.

## 1.- Between illusion and ignorance. Genesis of a *habitus* (1980s)

The first PHC nurses in Spain were entering a new field of work, for which they had no specific knowledge and therefore no special expectations or motivation. They arrived at PHC through an entrance exam, for two main reasons: the need to work or by chance.

They immediately detected their lack of knowledge and a great need to learn new things:

*When work began in the centres, it was a new world where there were no guidelines either. What we saw was that there was a field that had yet to be invented and explored.* (I4)

Along with this need to learn, another need arose: the need to unite among themselves and form a team. Everyone was in a similar situation of lack of knowledge and, therefore, the results depended on themselves. A team was formed in which the participation of everyone was key, and which provided "the illusion" of future possibilities and expectations.

After the team was formed, the need to find a guide arose. The leadership was assumed by a nurse of recognized prestige in the hospital environment, but who was also unaware of the new model of PHC, and therefore considered herself to be at the same level as her colleagues. In these initial moments, there was already talk of a "struggle" to define her competencies. This situation facilitated the lack of a vertical hierarchical model, as it was rather a horizontal and participatory leadership. Precisely for this reason, it became consolidated.

In this initial period, a structural element stands out that had an important symbolic value: the nurse's office, an exclusive space for each nurse to carry out her task. This space gave independence to each nurse by separating her physically from the doctor and meant that nurses assumed a greater responsibility, because they had to demonstrate the content of their work for which they deserved to have their own space, which also provided them with an identity. Similarly, nurses began to wear a white lab coat for greater comfort, however, this provoked some "confrontations" because until then it was only for doctors.

In addition, PHC nurses had to perform new work, acquire new skills and new knowledge, which meant a certain level of competition with other professionals, and in some cases, conflict.

> *And then, the confrontation with certain doctors who did not want their patients to see a nurse because, according to them, we were assuming the role of a doctor.* (I1)

Moreover, the population were unaware of what a PHC centre was, and what the role of the nurses was. Informants reported that, at the beginning, the patients valued them for supporting the doctor's work, and performed nursing techniques efficiently, therefore, when performing a different role, in which there was no dependence on the doctor or so many clinical techniques, they had to make an effort to obtain legitimacy, both from their patients and from other professionals.

The tension between strengths and constraints in this first stage forced nurses to seek and define their new role, thus the development of differentiated practices in the hospital environment and thereby, a particular *habitus* in PHC.

## 2.- The recognisable and recognised *habitus* (1990s)

Nursing professionals showed an important capacity of initiative to occupy their time in the consultation with new skills: assessment, monitoring and control of patients with chronic disease, development of protocols, and especially, the beginning of community work. Thus, the new role was strengthened. The informants described their efforts to define this role, seeking and claiming their own knowledge, which was increasingly broad and solid, and which gradually gave them more autonomy. The effort to continue learning and seek training to expand their knowledge was continuous.

> [Cardiac Auscultation] *"How fast", I don't seek anyone's opinion, I do the EKG and go to the doctor with the EKG and say: "Hey, he's in fibrillation". You have to know what fibrillation is. Or when you see something strange, what side effects can be caused by what he's taking, for example, a beta-blocker, because he has forty heartbeats. You have to have some in-depth knowledge of that pathology* (I1)

Social recognition became increasingly clear, because the population comes to their nurse demanding something, different from what the doctor provides: they were already becoming a reference for the population. Additionally, nurses continued to develop strategies to minimise conflicts with other professionals, who see in their progress a risk of losing competencies. Dialogue became an essential tool to address these conflicts.

In this phase, the first health care teams were also formed, consisting of a doctor and a nurse, under equal conditions, which required a negotiation process between the two, more or less explicit, to establish areas of work. Likewise, the nursing team continued to be strengthened, taking on the role of ensuring achievements are made, and the role of the leader continues to be relevant.

Nurses now became aware that they are offering a unique product, that they carry out their work without being subordinated or dependent on others and, therefore, with autonomy. Within their duties, nurses give special importance to the clinical assessment of their patients in a comprehensive manner, with a holistic vision. This work provides them with the trust and appreciation of the population, who recognise that nurses "improve" their health.

> *Then I always say*: *"Green head"* [the green colour in the computerized nursing record appears when the patient has already been assessed]. *And if you don't know that person who comes in and it's red, you have to fill it all out*: *toxic habits, food, weight and height if he or she is obese, a blood pressure test if it's lacking, and a blood test if they don't have one.* (I1)

However, while nurses are aware of the importance and complexity of their work, it is difficult for them to specify everything they do. They devote themselves to a multitude of tasks, with the risk that part of their work will be blurred, not evaluated, and a certain invisibility will persist. In any case, nurses are familiar with the new health model and believe in it; and the nursing role is consolidating. The *habitus* is confirmed.

## 3.- *Habitus* called into question (From 2000)

The health care model has demonstrated its effectiveness. However, contextual elements begin to emerge which tend to destabilize it. These elements are primarily political, and changes in the population's health profile.

Concerning the political elements, the successive changes in national and regional government administration, meant that each government proposed new guidelines in the health system that are ultimately reduced to statistical data, often far from the real needs of the population.

> *At that time I had the feeling that they were interested in percentages*: *"And how many diabetics do you have?" Do you remember those meetings? The number mattered more than the quality.* (I3)

In addition, decisions were unstable due to the many political changes, which add to the lack of continuity. An example of this is the "service portfolio", a variable offer in health care to the population, which sometimes fails to match the real possibilities and needs. Also, the cuts in staffing to decrease spending, reducing the ratio of nurses to users/patients.

In this context, the role of the leader, who guides and strengthens the team, is converted to a mere executor of the policy directions received, focused on managing these changes, minimising their impact, and readjusting the system.

The other novel element are changes in the profile of the population, in particular, the increase in aging and dependence. These are factors that impact nursing care, in addition to the incorporation of computer tools. The computerisation of health care brings advantages in the recording of the work performed, helping to quantify and measure work, however, together with the changes in the context, it blurs the professional role in community work.

In this situation, professionals begin to lose motivation, "locking themselves in" to their practice, doing what they feel like at each moment, a situation that generates inequality in the performance of the role and damages teamwork. There is no shared work methodology, and the visibility of the nursing role is reduced. The habitus now appears to be disintegrating.

*You get to a point where things get stagnant, both institutionally as well as at the group level.* (I9)

Informants are well aware of this situation and how it affects their professional role. While describing it, they were able to identify solutions, which focus on two areas: enhancing the evaluation of results of their work, and greater visibility for their duties. Both solutions are interlinked and could be achieved by improving it support for more efficient recording of the enormous complexity of nursing activities, and by promoting work with better methodology and scientific rigour.

## The central category: Autonomy

From the process of analysis and comparison between the main categories, a broader category emerged, Autonomy, which was implicitly or explicitly recognised in all other categories, and is shown as the guiding thread of the discourses. This is an autonomy that acquires its own characteristics in PHC: the capacity to decide one's training, to assume one's own leadership, to configure teams, and to acquire independent competencies.

> *There are many things that were really assumed by the nurses, the whole part of following up on chronic illnesses, the whole educational part, going out into the community, all of which we had to organise ourselves because no one was helping us.* (I1).

The participants pointed out that autonomy is especially consolidated by having the capacity to organise work in a way that is considered more efficient, allowing them to carry out the initiatives they consider appropriate with the population, thus reaching high levels of development and professional satisfaction.
*I think that the greatest satisfaction comes when people recognise that they have a nurse who handles things for them, that they can turn to her.* (I1)
In short, the opportunity that the conquest of autonomy represents serves as a motivation for complex work.

> *I have always felt autonomous in my work and I have always felt respected.* (I4)

When the professional role appears to falter, there is a loss in the ability to make decisions, a loss of autonomy, which is imposed by political or sociodemographic conditioning factors. However, for exercising the autonomy provided by PHC, it is necessary not only to be recognized by the population, but also by other professionals. This recognition is what leads to the legitimacy of the new role. Autonomy was sometimes also perceived as the independence to isolate oneself in one's own practice and make a difference in a more personal way.

> *For example, I look at my fellow primary care nurses and see the independence they have at work and how they handle things. I haven't even known that in my wildest dreams.* (I10)

When autonomy begins to disintegrate, it is difficult to reverse that process, which again becomes a new challenge.

> *I am talking about autonomy and self-management. We are aware that, in order to reach self-management, one must first experience autonomy, and this is a change that, unfortunately, has been going on for a few years and has been nipped in the bud. Recovering that is brutal.* (I11)

According to the participants in this study, sometimes, autonomy may be a reason for conflict with other professionals, although the majority, very naturally follow the initiatives of the

nurses within the team. Nurses who are subjected to this conflict run the risk of abandoning some practices to avoid confrontations. In contrast, those who are more integrated in the model, subordinate the conflict to the practice, recognising the value of autonomy over good personal relationships.

> *Are you asking me if I think our work is autonomous? I am absolutely convinced that it is, although some fellow doctors think it isn't and in fact strongly point this out, but I have the perception that nursing is a profession that collaborates with the medical profession, like the midwife, like with the social worker or like the pharmacist, but that it is autonomous.* (I12)

## Discussion

The participants in this study have helped identify the process by which the "search for autonomy" is the substantive theory that has generated the professional role in Spanish PHC nurses as a *habitus*. Autonomy manifests itself as an identity concept, resulting in the core category.

In the international context, several authors have analysed the autonomy of nurses in PHC [4,16,17,37–40] often identifying the "limits" or barriers they encounter to their development. However, in the case of the participants in our study, we found that the autonomous role is defined mainly in the nurses' work with the population, which enables them to seek a certain effectiveness and efficiency. Thus, the focus is on their actions and goals, mentioning the barriers in a secondary manner.

However, we agree with the previously cited authors that the limits to nurses' autonomy in PHC are those imposed on them by other professionals (mainly doctor) and those resulting from political decisions (lack of legal regulation, or influence of partisan interests) which, in our case, are also pointed out by informants as a threat that jeopardises the role.

According to our findings, the autonomous role in PHC nursing has its own characteristics that we have identified as the four elements that not only define it, but also contribute to its achievement (**Fig 1**): the need to acquire specific knowledge, the existence of a consolidated nursing team, strong leadership and social recognition. Of these, knowledge, linked to the

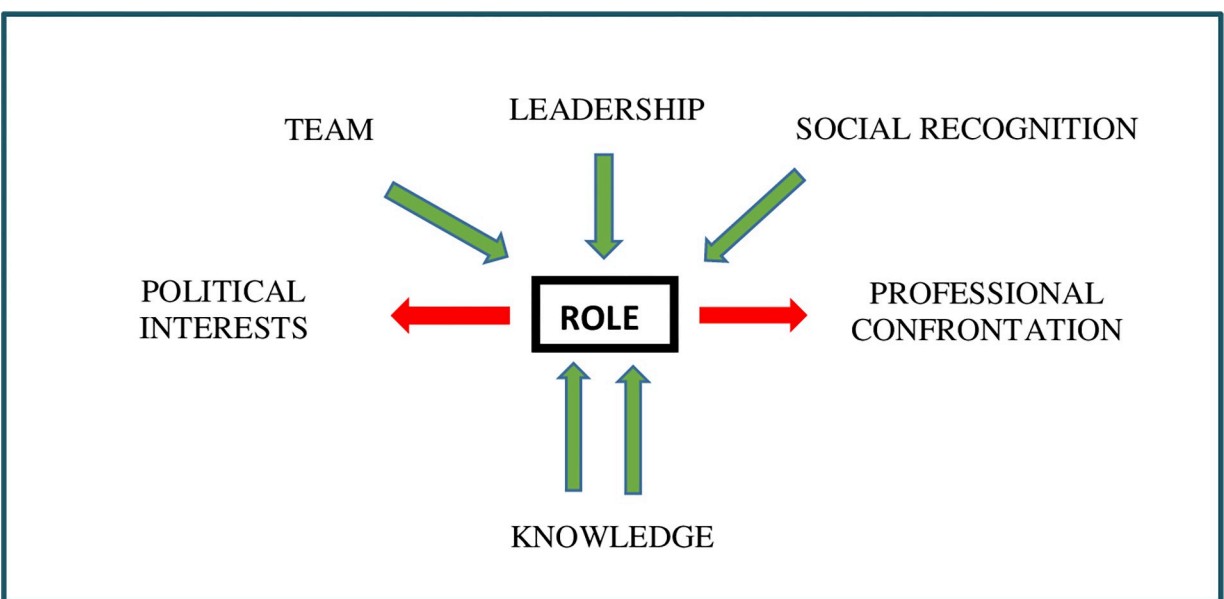

**Fig 1. Elements that shape the role.**

achievement of new competencies, is the key element to achieve the legitimisation and consolidation of the role.

We agree with authors [15,39] who point out the importance of acquiring specific knowledge in PHC. Other researchers specify that this knowledge should be acquired through regulated training, for example, as advanced practice [8,41,42] or the Community Nursing specialty which already exists in Spain. Specialised training plays an essential role in consolidating the role, which is why we propose it as one of the main objectives for those who manage PHC.

Some authors point to the possibility that the role may include interchangeable tasks between doctors and nurses [13,16,43,44], even with nurses replacing doctors [5,45]. This possibility suggests that there are ambivalent tasks, and would explain the possible conflicts we have pointed out with doctors, especially in the follow-up and control of patients with chronic conditions [17,39].

Furthermore, two elements have been identified that are detrimental to the configuration and consolidation of the role (Fig 1): political interests that are far removed from the real needs of the population and healthcare professionals; and confrontation, mainly with other professionals. Paradoxically, in our work we have found that to be legitimate, the transition from "practice" to "professional role" depends not only on the recognition by the population served, but also of other professionals, especially physicians. It is timely and strategic, therefore, to maintain a good personal relationship and good functioning within the interdisciplinary team [13]. This is also important for our informants, who propose dialogue as one of the most effective strategies to achieve this. Other strategies would involve considering and managing the symbolic value of other elements, such as nursing consultations.

Likewise, we have established that, in the case of PHC nursing, continuous practice generates *habitus*; and that *habitus* in turn generates practice, illustrating the close relationship between *habitus* and professional role. This fact must be considered if decisions are to be made to redefine roles, that is, acting on concrete practices, a *habitus* can be generated and therefore, a new professional role.

In this sense, it has been effective for us to adopt Bourdieu's sociology as a reference framework, as pointed out by other authors [25,46] for its usefulness in analysing and understanding the whole process of genesis and evolution of the role, and for its predictive power in explaining the regularity of actions, which will allow us to propose recommendations for practice.

## Limitations

As is usually the case in qualitative Grounded Theory studies, it is difficult to generalise the results when the research has been carried out with a limited group of participants in a given location. However, the results obtained in this case have allowed us to arrive at a series of concepts that, because of their level of abstraction beyond the concrete, can be useful in other contexts.

Moreover, two of the researchers knew and had experience in PHC, making it difficult to detach themselves from the object of study. We have attempted to overcome this limitation by discussing the process with the rest of the research team, who were not familiar with this field, which has made it possible to control possible biases.

## Conclusion

Since the implementation of the PHC model in Spain, the process of genesis and evolution of the nursing role has been marked by the search for professional autonomy.

An autonomous professional role exists when two elements are identified: the acquisition of *habitus*, through the performance of regular practices; and the recognition of this *habitus* by the population and other professionals. Both elements serve to legitimise the role.

The theoretical framework proposed by Bourdieu has been very useful in this study. Its prognostic capacity is operational to make the following recommendations for practice: the importance of strengthening the elements that influence the autonomy of the professional role (specialised knowledge, team, leadership, and social recognition), and avoiding those that harm that role (decisions guided by political interest and confrontation with other professionals).

## Supporting information

**S1 Table. Interview script (Spanish).**
(DOCX)

## Acknowledgments

We would like to thank Isabel Quintero, scientific translator and editor, for her assistance with the translation and proofreading of the article.

## Author Contributions

**Conceptualization:** Cristina Blanco-Fraile, María Madrazo-Pérez, Victor Fradejas-Sastre, Esperanza Rayón-Valpuesta.

**Data curation:** Cristina Blanco-Fraile, María Madrazo-Pérez, Victor Fradejas-Sastre, Esperanza Rayón-Valpuesta.

**Formal analysis:** Cristina Blanco-Fraile, María Madrazo-Pérez, Victor Fradejas-Sastre, Esperanza Rayón-Valpuesta.

**Funding acquisition:** Cristina Blanco-Fraile, María Madrazo-Pérez, Victor Fradejas-Sastre, Esperanza Rayón-Valpuesta.

**Investigation:** Cristina Blanco-Fraile, María Madrazo-Pérez, Victor Fradejas-Sastre, Esperanza Rayón-Valpuesta.

**Methodology:** Cristina Blanco-Fraile, María Madrazo-Pérez, Victor Fradejas-Sastre, Esperanza Rayón-Valpuesta.

**Project administration:** Cristina Blanco-Fraile, María Madrazo-Pérez, Victor Fradejas-Sastre, Esperanza Rayón-Valpuesta.

**Resources:** Cristina Blanco-Fraile, María Madrazo-Pérez, Victor Fradejas-Sastre, Esperanza Rayón-Valpuesta.

**Software:** Cristina Blanco-Fraile, María Madrazo-Pérez, Victor Fradejas-Sastre, Esperanza Rayón-Valpuesta.

**Supervision:** Cristina Blanco-Fraile, María Madrazo-Pérez, Victor Fradejas-Sastre, Esperanza Rayón-Valpuesta.

**Validation:** Cristina Blanco-Fraile, María Madrazo-Pérez, Victor Fradejas-Sastre, Esperanza Rayón-Valpuesta.

**Visualization:** Cristina Blanco-Fraile, María Madrazo-Pérez, Victor Fradejas-Sastre, Esperanza Rayón-Valpuesta.

**Writing – original draft:** Cristina Blanco-Fraile, María Madrazo-Pérez, Victor Fradejas-Sastre, Esperanza Rayón-Valpuesta.

**Writing – review & editing:** Cristina Blanco-Fraile, María Madrazo-Pérez, Victor Fradejas-Sastre, Esperanza Rayón-Valpuesta.

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
