## [Decision Letter · Decision Letter 0]

17 Mar 2021

PONE-D-21-01697

The role of nursing in primary health care using Bourdieu’s concept of habitus. A grounded theory study.

PLOS ONE

Dear Dr. Víctor Fradejas-Sastre,

Thank you for submitting your manuscript to PLOS ONE. After careful consideration, we feel that it has merit but does not fully meet PLOS ONE’s publication criteria as it currently stands. Therefore, we invite you to submit a revised version of the manuscript that addresses the points raised during the review process.

Thank you for your submission of this manuscript to PLOS ONE. The paper addresses an interesting and important topic and provides novel information in the area of primary healthcare nursing roles. Detailed comments are provided below, both from the reviewer and the Editor's evaluation. All comments should be addressed. 

In providing your response, in addition to the requirements below, also ensure that your manuscript has page numbers and line numbers for ease of review. 

We look forward to receiving your revised manuscript.

Kind regards,

Nelly Oelke

Academic Editor

PLOS ONE

Journal Requirements:

3. Please ensure that you include a title page within your main document. We do appreciate that you have a title page document uploaded as a separate file, however, as per our author guidelines (http://journals.plos.org/plosone/s/submission-guidelines#loc-title-page) we do require this to be part of the manuscript file itself and not uploaded separately.

4. Please include a copy of the interview guide used in the study, in both the original language and English, as Supporting Information, or include a citation if it has been published previously."

5. Thank you for stating in the text of your manuscript "This research was approved by the Health Service of Cantabria, according to the Declaration of Helsinki (World Medical Association, 2013). The informed consent of all the participants included in the study was requested, informing them of the voluntary nature of participation and the possibility of withdrawing from the study at any time. The data obtained were treated to guarantee confidentiality and anonymity, according to the Law on Protection of Personal Data in force in Spain (Law 15/1999)."

Please verify whether the ethics committee specifically approved your study.

Please also state what type of consent you obtained (for instance, written or verbal, and if verbal, how it was documented and witnessed).

Please also add all of this information to your ethics statement in the online submission form."

Additional Editor Comments (if provided):

Overall comments: This is both an interesting and important topic. The paper is overall well written and organized. You use the phrase “in order” a fair bit; would recommend deleting the same or at least a number of them as they are extraneous words and don’t really add anything.

Introduction:

• I would argue that PHC is the not the first point of contact with only national health care systems.

• “has meant a new model…” – would suggest revising “has meant” to something else as it does not flow well, and is confusing. Perhaps used “was defined as”, or “was established as.”

• First sentence in the last paragraph, seems a bit confusing. Would also suggest dividing into two sentences.

Sample/Participants:

• In your inclusion criteria it isn’t really clear who participated in interviews. Nurses? Nurse managers?

Results:

• “In addition, PHC nurses had to perform new work, acquire new skills and new knowledge, which meant a certain level competition with other professionals, and in some cases, conflict.” Please add “of” to “certain level of competition.” Also, this sentence is important sentence but not supported in your writing (single sentence in a paragraph). Could you add a quote, or add additional description?

• Section 2 – “chronic patients” should be “patients with chronic conditions” or “disease”

• Section 2, in particular, is a bit choppy. It has a lot of different paragraphs and often just containing a single sentence. Combine paragraphs or add sentences to further describe and increase flow.

• Section 3, paragraph 2 -the word “relays” is used. Could you describe more? In of itself, it’s meaning is not clear.

• Section 3, paragraph 5 – “impact on nursing care” remove “on.” Also remove “another instrumental factor:”

• Last paragraph in Section 3 – further develop the recommendations from informants.

• Autonomy section – also has a lot of paragraphs and a number of them with only a single sentence. As above, consider combining or adding description as needed.

Discussion:

• Figure 1 – you also include political interests and professional confrontation in the figure with not description of these in the discussion section. Please add description. It is also interesting that these are outgoing arrows, whereas these I think actually impact the role quite significantly.

• Paragraph 6 – “chronic patients” – see previous comment.

Table 1:

• Is potentially quite identifiable. I would delete the same and add a description in the text, ensuring that details cannot identify persons involved in the interviews.

Reviewers' comments:

Reviewer's Responses to Questions

**Comments to the Author**

1. Is the manuscript technically sound, and do the data support the conclusions?

Reviewer #1: Yes

2. Has the statistical analysis been performed appropriately and rigorously? 

Reviewer #1: N/A

3. Have the authors made all data underlying the findings in their manuscript fully available?

Reviewer #1: Yes

4. Is the manuscript presented in an intelligible fashion and written in standard English?

Reviewer #1: Yes

5. Review Comments to the Author

Reviewer #1: Thank you for the opportunity to review this manuscript. This paper presents a very interesting application of a sociological model to understand the evolution of the nursing role in primary health care. This application is very novel and although only 1 country was investigated, the content and findings of this study can be easily applicable to nurses from a wide array of geographic locations. Below are some suggestions to strengthen the manuscript for clarity of some of the content.

INTRODUCTION/Background

It remains unclear whether the targeted group of nurses in the study are traditional registered nurses (RNs) or advanced practice registered nurses (APRNs) such as nurse practitioners. There is an increased amount of evidence emerging that differentiate between RNs and APRNs in primary care. Therefore I think it is important to clarify this early on. There may be overlap in these roles over time but for the purpose of implementing the recommendations at the end, it should be clear. (Citations used to support the role of nurses in primary care include both RNs and APRNs). Regardless, the findings of the paper are easily applicable to either type of primary care nurse and can used as such.

4th paragraph “The arrival of the new professionals was conditioned by the dominant models” This sentence should be explained more in detail. New professionals are the nurses and the dominant models are traditional physician-led care? If so, this should included to present the context to the reader.

Also in this paragraph, what is meant by “nursing role in PHC has fundamentally been built based on practice”? Why is this negative and presents a challenge? How has a role built on “practice” inhibited the nursing role and its definition?

Overall, the background section may benefit from a brief history of the scope of practice of existing nurses in Spain. It appears that there was a complete absence of nurses in primary care prior to 1980s and now has expanded. To better understand the context of the nursing role in these particular settings, a very brief explanation of the history of the nursing role in Spain may help further interpret the evolution described in the results.

METHODS

Rigorous and cited methodology was used to perform this study. Excellent description of the through qualitative methodology used.

Sample

Please clarify what is meant by “the first nurse manager and two other directors from different periods” What is meant by different periods?

Also, what is considered “extensive experience?”

Inclusion criteria is vague. What constitutes “being active in the different stages of PHC since its origin”? Does this equate to a long duration working in primary care since its inception? One role vs. several different roles over the years?

Data analysis

Last paragraph: “emerging concepts were ordered according to the timeline described by the informants, using a theoretical model.” Describe more in detail how the participant perspectives were aligned and coded using this timeline/model. Re-state the name of the model.

Results

The methods had described analysis using a timeline yet the results do not differentiate opinions based on stages or years. It would be very interesting to understand which quotes/evolving roles are aligned with different decades compared to current roles. This is especially important given that some participants may have been in primary care longer than others.

Discussion

Paragraph 7 Change “depends not only on the recognition OF the population served” to “recognition BY the population served.”

Title: After reading the paper, I think it would benefit the authors to include “evolution” somewhere in the title since the results and primary aims of this study was understand the evolution and perspectives of role of primary care nurses in Spain theoretically.

In summary, I thoroughly enjoyed reading this paper. This is novel evidence and very innovative compared to current literature that is emerging about primary care nurses conceptually. The results and recommendations stemming from this paper will have wide implications for policy, practice, theory and future research. Well done.

6. PLOS authors have the option to publish the peer review history of their article (what does this mean?). If published, this will include your full peer review and any attached files.

Reviewer #1: **Yes: **Allison A. Norful, PhD, RN, ANP-BC

---

## [Author Response · Author response to Decision Letter 0]

20 Jul 2021

Nelly Oelke

Academic Editor

PLOS ONE

May 1, 2021

Dear Dr. Nelly Oelke,

Thank you for the opportunity to revise our manuscript, The evolution of the role of nursing in primary health care using Bourdieu’s concept of habitus. A grounded theory study. We appreciate the careful review and constructive suggestions. It is our belief that the resubmitted manuscript is substantially improved after making the suggested edits. 

Revisions in the text are shown using yellow highlight for additions and edits. The revision, based on the review team’s collective input, includes a number of positive changes. Based on your guidance in your email dated March 17, among other changes, we have: 

• Updated the literature review (including 1 article).

• Deleted table 1, as suggested by the Editor.

• Double checked author guidelines.

In the following pages please find a point-to-point response to each of the comments received. We hope that the revisions in the manuscript and our accompanying responses will be sufficient to make our manuscript suitable for publication in PLOS ONE. 

We shall look forward to hearing from you at your earliest convenience.

Yours sincerely,

Dr. Víctor Fradejas-Sastre

Corresponding Author

4RESPONSE TO ACADEMIC EDITOR AND REVIEWER

Comment 4. Please include a copy of the interview guide used in the study, in both the original language and English, as Supporting Information, or include a citation if it has been published previously.

Response: We have now included the interview guide as requested in Table 1, in both the original language (Spanish) and in English.

Comment 5. Thank you for stating in the text of your manuscript "This research was approved by the Health Service of Cantabria, according to the Declaration of Helsinki (World Medical Association, 2013). The informed consent of all the participants included in the study was requested, informing them of the voluntary nature of participation and the possibility of withdrawing from the study at any time. The data obtained were treated to guarantee confidentiality and anonymity, according to the Law on Protection of Personal Data in force in Spain (Law 15/1999)."

Please verify whether the ethics committee specifically approved your study.

Response:

The Health Service of Cantabria specifically approved this study, verbally authorising access and contact with the informants, through the Director of Nursing at that time. The study was authorised by the Primary Care Directorate of the Health Service as they were aware that written informed consent would be requested from each of the informants and that all the ethical requirements of the Declaration of Helsinki would be met. 

We have rewritten this paragraph to more clearly reflect this information and in line with the written documentation available, which we are making available to the editor.

The text has been edited as follows, pg. 7:

“This research was conducted according to the Declaration of Helsinki (World Medical Association, 2013). The informed consent of all the participants included in the study was requested and provided in writing, informing participants of the voluntary nature of their participation and the possibility of withdrawing from the study at any time. The data obtained were treated to guarantee confidentiality and anonymity, according to the Law on Protection of Personal Data in force in Spain (Law 15/1999).”

Please also state what type of consent you obtained (for instance, written or verbal, and if verbal, how it was documented and witnessed).

Response: Informed consent was in writing. We have documented and collected all relevant documents from each of the participants. The model of informed consent form is attached below. For reasons of space, it is not included in the Article for publication. If it is necessary to include it as an Annex, please do not hesitate to let us know.

Please also add all of this information to your ethics statement in the online submission form."

Response: Done

ADDITIONAL EDITOR COMMENTS:

Overall comments: This is both an interesting and important topic. The paper is overall well written and organized. You use the phrase “in order” a fair bit; would recommend deleting the same or at least a number of them as they are extraneous words and don’t really add anything.

Response: we have deleted the expression “in order” from the following lines: 91, 93, 128, 391, 425.

Introduction:

• I would argue that PHC is the not the first point of contact with only national health care systems.

Response: We have changed the wording to a more appropriate expression: PHC "as the first level of health services delivery". Lines 36 and 37.

The text has been edited as follows, pg. 2:

“…primary health care (PHC) as the first level of health service delivery.”

• “has meant a new model…” – would suggest revising “has meant” to something else as it does not flow well, and is confusing. Perhaps used “was defined as”, or “was established as.”

Response: We have followed this suggestion in line 37. 

• First sentence in the last paragraph, seems a bit confusing. Would also suggest dividing into two sentences.

Response: We have changed the wording to make it more understandable and split the sentence into two parts.

The text has been edited as follows, pg. 3:

“The arrival of the new professionals, both RNs and doctors, was conditioned by the dominant models taken from hospital care, with a focus on pathology and with a tendency towards curative care, in accordance with their training oriented to addressing the immediate patient needs (Lamata, Pérez, 2011). This has meant that throughout these almost 40 years, the RN role in PHC has fundamentally been built based on practice, i.e., learning by doing. The new role of nursing in PHC was proposed under broad, unspecific lines, however, the population's need for professional care was growing exponentially and was not only based on the immediacy of care. In the absence of a model or reference guide, RNs have been constructing the new role as they work, based on trial and error. To be able to analyse this role and overcome these new challenges, the present study aims to determine the process, from a holistic viewpoint, through which Spanish nurses have developed a differentiated role in PHC”.

Sample/Participants:

• In your inclusion criteria it isn’t really clear who participated in interviews. Nurses? Nurse managers?

Response: Both basic nurses and nurse managers participated. We rewrote the inclusion criteria to specify this in lines 122, 123, and 133. 

Results:

• “In addition, PHC nurses had to perform new work, acquire new skills and new knowledge, which meant a certain level competition with other professionals, and in some cases, conflict.” Please add “of” to “certain level of competition.” 

Response: change made in page 226.

• Also, this sentence is important sentence but not supported in your writing (single sentence in a paragraph). Could you add a quote, or add additional description?

Response: we have added an informant text transferred from Table 3. ADDITIONAL PARTICIPANT NARRATIVES

• Section 2 – “chronic patients” should be “patients with chronic conditions” or “disease”

Response: We have changed this in lines 82, 241 and 386.

• Section 2, in particular, is a bit choppy. It has a lot of different paragraphs and often just containing a single sentence. Combine paragraphs or add sentences to further describe and increase flow.

Response: We have corrected this.

• Section 3, paragraph 2 -the word “relays” is used. Could you describe more? In of itself, it’s meaning is not clear.

Response: We refer to the various changes of government administration in the region and at the national level.

The text has been edited as follows, pg. 13:

“Concerning the political elements, the successive changes in national and regional government administration , meant that each government proposed new guidelines in the health system that are ultimately reduced to statistical data, often far from the real needs of the population. “

• Section 3, paragraph 5 – “impact on nursing care” remove “on.” Also remove “another instrumental factor:”

Response: we have removed this in lines 295 and 296.

• Last paragraph in Section 3 – further develop the recommendations from informants.

Response: Further specification of these recommendations has been added. We hope that this will be sufficient.

The text has been edited as follows, pg. 14:

“Informants are well aware of this situation and how it affects their professional role. While describing it, they were able to identify solutions, which focus on two areas: enhancing the evaluation of results of their work, and greater visibility for their duties. Both solutions are interlinked and could be achieved by improving it support for more efficient recording of the enormous complexity of nursing activities, and by promoting work with better methodology and scientific rigour.”

• Autonomy section – also has a lot of paragraphs and a number of them with only a single sentence. As above, consider combining or adding description as needed.

Response: We have reduced the number of paragraphs.

Discussion:

• Figure 1 – you also include political interests and professional confrontation in the figure with not description of these in the discussion section. Please add description. It is also interesting that these are outgoing arrows, whereas these I think actually impact the role quite significantly.

Response: We have included a description of these elements in the Discussion. Lines 388-391.

The text has been edited as follows:

“Furthermore, two elements have been identified that are detrimental to the configuration and consolidation of the role (Figure 1): political interests that are far removed from the real needs of the population and healthcare professionals; and confrontation, mainly with other professionals”.

• Paragraph 6 – “chronic patients” – see previous comment.

Response: corrected. Line 386.

Table 1:

• Is potentially quite identifiable. I would delete the same and add a description in the text, ensuring that details cannot identify persons involved in the interviews.

Response: Table 1 has been deleted, and the numbering of the other tables in the text has been corrected. We have expanded the description of the sample in the corresponding section. 

REVIEWERS' COMMENTS: REVIEWER'S RESPONSES TO QUESTIONS

INTRODUCTION/Background

* It remains unclear whether the targeted group of nurses in the study are traditional registered nurses (RNs) or advanced practice registered nurses (APRNs) such as nurse practitioners. There is an increased amount of evidence emerging that differentiate between RNs and APRNs in primary care. Therefore I think it is important to clarify this early on. There may be overlap in these roles over time but for the purpose of implementing the recommendations at the end, it should be clear. (Citations used to support the role of nurses in primary care include both RNs and APRNs). Regardless, the findings of the paper are easily applicable to either type of primary care nurse and can used as such.

Response: We have specified which type of nurses we are referring to in lines 47; 54; 66, and 67. We explain this situation of Spanish PHC nurses in lines 77-79.

*4th paragraph “The arrival of the new professionals was conditioned by the dominant models” This sentence should be explained more in detail. New professionals are the nurses and the dominant models are traditional physician-led care? If so, this should included to present the context to the reader.

Response: It has been rewritten as follows, providing more details:

“The arrival of the new professionals, both RNs and doctors, was conditioned by the dominant models taken from hospital care, with a focus on pathology and with a tendency towards curative care, in accordance with their training oriented to addressing the immediate patient needs (Lamata, Pérez, 2011). This has meant that throughout these almost 40 years, the RN role in PHC has fundamentally been built based on practice, i.e., learning by doing. The new role of nursing in PHC was proposed under broad, unspecific lines, however, the population's need for professional care was growing exponentially and was not only based on the immediacy of care. In the absence of a model or reference guide, RNs have been constructing the new role as they work, based on trial and error. To be able to analyse this role and overcome these new challenges, the present study aims to determine the process, from a holistic viewpoint, through which Spanish nurses have developed a differentiated role in PHC”

*Also in this paragraph, what is meant by “nursing role in PHC has fundamentally been built based on practice”? Why is this negative and presents a challenge? How has a role built on “practice” inhibited the nursing role and its definition?

Response: The meaning of this sentence has been explained in the previous section and in the text on lines 58-62.

*Overall, the background section may benefit from a brief history of the scope of practice of existing nurses in Spain. It appears that there was a complete absence of nurses in primary care prior to 1980s and now has expanded. To better understand the context of the nursing role in these particular settings, a very brief explanation of the history of the nursing role in Spain may help further interpret the evolution described in the results.

Response: A short history has been included by adding a new paragraph as follows:

“In Spain, before the 1980s, there were two types of care provided outside the hospital setting: in outpatient clinics and in rural areas (Martín, A., Ledesma, A., & Sans, A. 2000). In outpatient clinics, consultations were largely devoted to administrative procedures, such as obtaining sick leave, tasks that were often delegated to the nurse who accompanied the doctor in the consultation. Sick people were only attended to when they came to the doctor. In rural areas, care was provided at home. In both cases there was a big difference with the hospital environment, which was more financially endowed and more prestigious. Morever, the nurses were all RNs, as there was no APRN and the specialty of Community Nursing was approved in the year 2012”.

METHODS

Rigorous and cited methodology was used to perform this study. Excellent description of the through qualitative methodology used.

Sample

*Please clarify what is meant by “the first nurse manager and two other directors from different periods” What is meant by different periods?

Response: We refer to the fact that we interviewed the first nurse manager who started the new PHC model, and also two subsequent directors of nursing, who carried out their tasks as managers in different periods of time. It was important to collect the testimonies of people who had joined the model at different points in its evolution since 1986. This has been added to the text for clarity. Lines 127-130: 

“The first nurse manager and two other later directors from different periods all of whom were interviewed to collect testimonies from managers who had joined the model at different points in its evolution since 1986. Most of the participants had over 10 years of experience in PHC.”

*Also, what is considered “extensive experience?

Response: More than 10 years in PHC. We have modified the wording to include this specific data.

*Inclusion criteria is vague. What constitutes “being active in the different stages of PHC since its origin”? Does this equate to a long duration working in primary care since its inception? One role vs. several different roles over the years?

Response: This refers to the fact that the participants were still working in PHC at the time of the interview, and that their work covered different periods of years, some since their inception, fulfilling both staff nurse and nurse manager roles. We have added this to the text to clarify these ideas in lines 131 -133 as follows:

“The inclusion criteria were: voluntary participation in the study, being active, i.e. still working in PHC at the time of the interview; some for a long period of time in the different stages of PHC since its origin, and who would perform both staff nurse and nurse manager roles”.

Data analysis

*Last paragraph: “emerging concepts were ordered according to the timeline described by the informants, using a theoretical model.” Describe more in detail how the participant perspectives were aligned and coded using this timeline/model. Re-state the name of the model.

Response: The timeline was developed as the research progressed. The participants' accounts made it possible to identify three time periods, differentiated from each other by the content of the narratives that marked a change of era, and which were named according to their own content.

The "model" is a diagram inspired by what Strauss and Corbin call the "conditional/consequential matrix". The diagrams were developed from the beginning of the research in each of the 3 time periods, in order to reflect the relationships of the concepts to each other and to facilitate coding and categorisation. Some of these diagrams are attached here, but due to the complexity of their explanation, we consider that they would exceed the length required for this article. 

This paragraph has been reworded to make it clearer, as follows:

“From the initial stages of this study the emerging concepts were ordered according to the timeline described by the informants, which made it possible to identify different periods of time, differentiated from each other according to the content of their narratives, and which were marking a change of era A theoretical framework was used in the form of a diagram, inspired by the conditional/consequential matrix (Strauss and Corbin, 2008), to explain the relationships of the concepts among each other for each time period”.

DIAGRAM 1

DIAGRAM 2

DIAGRAM 3 

Results

*The methods had described analysis using a timeline yet the results do not differentiate opinions based on stages or years. It would be very interesting to understand which quotes/evolving roles are aligned with different decades compared to current roles. This is especially important given that some participants may have been in primary care longer than others.

Response: We have added the approximate period of time corresponding to each of the 3 sections indicated in the Results. Although some participants have been in PHC longer than others, their discourses have been included in the period corresponding to the content of their narrative, for example in the case of Informant No. 1. We have paid attention to the narratives that they themselves placed in each period, which has allowed us to establish the coding and categorisation in each of the different periods.

Discussion

*Paragraph 7 Change “depends not only on the recognition OF the population served” to “recognition BY the population served.”

• - We have made the suggested change. Line 392

Title: 

*After reading the paper, I think it would benefit the authors to include “evolution” somewhere in the title since the results and primary aims of this study was understand the evolution and perspectives of role of primary care nurses in Spain theoretically.

Response: We have made the suggested change.

---

## [Decision Letter · Decision Letter 1]

10 Nov 2021

PONE-D-21-01697R1The role of nursing in primary health care using Bourdieu’s concept of habitus. A grounded theory study.PLOS ONE

Dear Dr. Fradejas-Sastre,

Thank you for submitting your manuscript to PLOS ONE. After careful consideration, we feel that it has merit but does not fully meet PLOS ONE’s publication criteria as it currently stands. Therefore, we invite you to submit a revised version of the manuscript that addresses the points raised during the review process.

We look forward to receiving your revised manuscript.

Kind regards,

César Leal-Costa, Ph. D

Academic Editor

PLOS ONE

Journal Requirements:

Reviewers' comments:

Reviewer's Responses to Questions

**Comments to the Author**

1. If the authors have adequately addressed your comments raised in a previous round of review and you feel that this manuscript is now acceptable for publication, you may indicate that here to bypass the “Comments to the Author” section, enter your conflict of interest statement in the “Confidential to Editor” section, and submit your "Accept" recommendation.

Reviewer #2: All comments have been addressed

Reviewer #3: (No Response)

2. Is the manuscript technically sound, and do the data support the conclusions?

Reviewer #2: Yes

Reviewer #3: Yes

3. Has the statistical analysis been performed appropriately and rigorously? 

Reviewer #2: N/A

Reviewer #3: N/A

4. Have the authors made all data underlying the findings in their manuscript fully available?

Reviewer #2: Yes

Reviewer #3: Yes

5. Is the manuscript presented in an intelligible fashion and written in standard English?

Reviewer #2: Yes

Reviewer #3: Yes

6. Review Comments to the Author

Reviewer #2: After analyzing the article and the revisions proposed by the reviewers, it is necessary to indicate that the authors have been able to respond efficiently to the proposed revisions.

The use of the theoretical framework proposed by Bourdieu has been a success for the study of the evaluation of the role of the primary care nurse.

The introduction makes a general review of the study problem and presents the background in a global way leading the reader to the study objective. The inclusion of recommendations has improved this section.

The methodology is detailed and transparently exposes the design and process followed. In this sense, in the section on methodological rigor, it would be necessary to clarify certain aspects:

1- It is mentioned that maximum heterogeneity was sought in the sample. This is a good thing, since saturation is achieved in a sample with great variability. But it would be necessary to demonstrate this variability. It would be advisable to include a table with characteristics of the sample in which this variability is evidenced.

2- It is reported that the discussion within the team was facilitated at the time of the analysis and mention is made of triangulation. It should briefly detail what type of triangulation and how it was performed.

Reviewer #3: The paper reports on developing a theory grounded from the data generating How Spanish nurses acquire a role in PHC. This paper is potential for a publication. Nevertheless, the authors need to refine few sections of the paper to strengthen the paper.

Regarding the choice of this grounded theory design for this study, how do you define and describe your research position in this study?

Could you please explain how you maintain the study rigor? For example, when validating the result to the participants.

The authors mention about the second interview. It is necessary to state whether the interview was undertaken to existing participants or additional ones? please, explain.

Under the ethical consideration section, I suggest the authors state the number of the letter (of the ethics approval).

During the research study process, two languages (English and Spanish) have been used, thus the authors need to briefly explain the translation process. The readers need to know whether it was undertaken during analytical process or after the study was concluded (when the final theory is emerged).

Finally, the authors need to highlight the study originality that would help strengthen the paper's conclusions and contribution.

7. PLOS authors have the option to publish the peer review history of their article (what does this mean?). If published, this will include your full peer review and any attached files.

Reviewer #2: **Yes: **Ismael Jimenez Ruiz

Reviewer #3: No

---

## [Author Response · Author response to Decision Letter 1]

10 Feb 2022

Rebuttal letter

RESPONSE TO ACADEMIC EDITOR AND REVIEWER

Reviewer #2: 

The methodology is detailed and transparently exposes the design and process followed. In this sense, in the section on methodological rigor, it would be necessary to clarify certain aspects:

1- It is mentioned that maximum heterogeneity was sought in the sample. This is a good thing, since saturation is achieved in a sample with great variability. But it would be necessary to demonstrate this variability. It would be advisable to include a table with characteristics of the sample in which this variability is evidenced.

RESPONSE. In the first version submitted to PLOS ONE, we included a Table 1, which precisely detailed the characteristics of the sample. However, in the Additional Editor Comments, we received the following suggestion: "Table 1: Is potentially quite identifiable. I would delete the same and add a description in the text, ensuring that details cannot identify persons involved in the interview". 

For this reason, we then withdrew Table 1, and included the description in the text, as suggested. We leave it to the editors to decide whether or not to publish it. Our opinion is in agreement with what was suggested in Additional Editor Comments, since we have included several tables and figures, and perhaps it would be better to reduce their number as much as possible.

2- It is reported that the discussion within the team was facilitated at the time of the analysis and mention is made of triangulation. It should briefly detail what type of triangulation and how it was performed.

RESPONSE. We referred to researcher triangulation. As we explained in the manuscript, the principal investigator had extensive knowledge of PHC and this made it convenient for the research team to carry out a frequent exercise of reflection in order to control possible biases. This exercise among the researchers was applied during the coding and categorization process, and also in the interpretation of the results, through a second review by another researcher of the data proposed by the principal investigator, contrasting later those aspects within the team that were doubtful or even divergent. 

We appreciate this comment and include this detail in the text: 

“researcher triangulation, which was applied during the coding and categorization process, and also in the interpretation of the results, through a second review by another researcher of the data proposed by the main researcher. Any aspects that were doubtful or divergent were further discussed within the team”.

Reviewer #3: 

1.- Regarding the choice of this grounded theory design for this study, how do you define and describe your research position in this study?

RESPONSE. The decision to adopt Grounded Theory was made within the research team, after discussion and analysis of other possible options (such as the phenomenological approach and ethnography), which were discarded because the researchers, supported by the expert knowledge of the principal researcher on PHC, realized that in the whole process of configuration of the new role, an explanatory theory of this evolution could be found, and therefore, the decision was made to select the Grounded Theory design. 

The position of the principal investigator in this work was determined by her extensive knowledge of primary care, with constant contact over the years with its professionals.

To clarify this point, we include this information in the manuscript, in the section on data analysis, in the following terms: 

“which among other aspects, facilitated the decision to opt for Grounded Theory”. 

2.- Could you please explain how you maintain the study rigor? For example, when validating the result to the participants.

RESPONSE. Rigor was generally maintained, respecting the criteria indicated by Strauss & Corbin (2008) and Calderón, C., (2002). Specifically, the validation of the participants' results was carried out by returning them to the informants, that is, the results of the work were communicated to the person who conducted the pilot interview, as well as to the key informant, identifying and confirming both the coherence (credibility) with their own reality.

In the manuscript we have revised and edited the wording to make it clearer, as follows: 

“For this purpose, the results were also returned to the informants, that is, the results of the work were communicated to the nurse who conducted the pilot interview, as well as to the key informant, both of whom identified and confirmed the coherence (credibility) with their own reality”.

3.- The authors mention about the second interview. It is necessary to state whether the interview was undertaken to existing participants or additional ones? please, explain.

RESPONSE. Firstly, in order to achieve an approximation to the studied reality, a first interview or pilot interview was carried out, which was later treated like the rest of the data. For this first interview, a nurse was selected who started in PHC with the new model, by means of an exam and with practically no previous work experience. At the time of the interview, she was still working at the same Health Center. In addition, another nurse interviewed was considered a key informant, as she coincided with a diversity of circumstances that gave her a broad vision in relation to the contributions to the study. Both were participants included in the sample.

4.- Under the ethical consideration section, I suggest the authors state the number of the letter (of the ethics approval).

RESPONSE. Unfortunately, we are unable to provide a number of the approval letter because at the time the fieldwork was conducted, there was no registration number or code listed on that letter (in Cantabria region). 

5.- During the research study process, two languages (English and Spanish) have been used, thus the authors need to briefly explain the translation process. The readers need to know whether it was undertaken during analytical process or after the study was concluded (when the final theory is emerged).

RESPONSE. Only Spanish was used during the study. The manuscript was then translated into English by a scientific translation service for manuscripts.

Perhaps some of the expressions used may have caused the doubts raised by the reviewer, and therefore we will review the wording. 

6.- Finally, the authors need to highlight the study originality that would help strengthen the paper's conclusions and contribution.

RESPONSE. We appreciate this comment which will certainly strengthen this research. We have added the following paragraph in the Rigor section following this suggestion:

“and, in addition, through two fundamental aspects: first, the application of Bourdieu's theoretical framework to the practice of health care, and second, the results obtained, which illustrate in an operative manner the elements that favor and those that hinder the development of the professional role”.

---

## [Decision Letter · Decision Letter 2]

2 Mar 2022

The evolution of the role of nursing in primary health care using Bourdieu’s concept of habitus. A grounded theory study.

PONE-D-21-01697R2

Dear Dr. Fradejas-Sastre,

We’re pleased to inform you that your manuscript has been judged scientifically suitable for publication and will be formally accepted for publication once it meets all outstanding technical requirements.

Kind regards,

César Leal-Costa, Ph. D

Academic Editor

PLOS ONE

Additional Editor Comments (optional):

Reviewers' comments:

Reviewer's Responses to Questions

**Comments to the Author**

1. If the authors have adequately addressed your comments raised in a previous round of review and you feel that this manuscript is now acceptable for publication, you may indicate that here to bypass the “Comments to the Author” section, enter your conflict of interest statement in the “Confidential to Editor” section, and submit your "Accept" recommendation.

Reviewer #2: All comments have been addressed

Reviewer #3: All comments have been addressed

2. Is the manuscript technically sound, and do the data support the conclusions?

Reviewer #2: Yes

Reviewer #3: (No Response)

3. Has the statistical analysis been performed appropriately and rigorously? 

Reviewer #2: N/A

Reviewer #3: N/A

4. Have the authors made all data underlying the findings in their manuscript fully available?

Reviewer #2: Yes

Reviewer #3: Yes

5. Is the manuscript presented in an intelligible fashion and written in standard English?

Reviewer #2: Yes

Reviewer #3: Yes

6. Review Comments to the Author

Reviewer #2: The authors have responded to the specifications of the requested minor revision.

Therefore my decision is to accept the submitted manuscript in the latest revision version.

Reviewer #3: Dear Authors,

Thank you for the authors' response to my queries. Regarding the ethical approval, even though no letter of approval has been provided but the authors have explicated ethical considerations in their study. However, in the future research study, I suggest that any research studies involving human beings as participants need to be registered or applied to a human research ethic committee/board. If you do not have any HREC in the study setting/region, you may register it to the closest one to the region.

Overall, this study is publishable. Thank you.

Regards,

The reviewer.

7. PLOS authors have the option to publish the peer review history of their article (what does this mean?). If published, this will include your full peer review and any attached files.

Reviewer #2: **Yes: **Ismael Jiménez Ruiz

Reviewer #3: No

---

## [Editor Report · Acceptance letter]

7 Mar 2022

PONE-D-21-01697R2 

The evolution of the role of nursing in primary health care using Bourdieu’s concept of habitus. A grounded theory study. 

Dear Dr. Fradejas-Sastre:

I'm pleased to inform you that your manuscript has been deemed suitable for publication in PLOS ONE. Congratulations! Your manuscript is now with our production department. 

Kind regards, 

on behalf of

Dr. César Leal-Costa 

Academic Editor

PLOS ONE